# Meta-Analysis of European Clinical Trials Characterizing the Healthy-Adult Serum 25-hydroxyvitamin D Response to Vitamin D Supplementation

**DOI:** 10.3390/nu15183986

**Published:** 2023-09-14

**Authors:** Manuel Rupprecht, Stefan Wagenpfeil, Jakob Schöpe, Reinhold Vieth, Thomas Vogt, Jörg Reichrath

**Affiliations:** 1Department of Dermatology, Saarland University Medical Center, 66421 Homburg, Germany; 2Health Management, German University for Prevention and Health Management (DHfPG), 66123 Saarbruecken, Germany; 3Institute for Medical Biometry, Epidemiology and Medical Informatics, Saarland University Medical Center, 66421 Homburg, Germany; 4Department of Nutritional Sciences, Department of Laboratory Medicine and Pathobiology, Faculty of Medicine, University of Toronto, Toronto, ON M5G 1V7, Canada

**Keywords:** vitamin D, supplementation, Europe, vitamin D deficiency, adults, healthy

## Abstract

To obtain reliable data that allow health authorities to re-evaluate recommendations for oral vitamin D uptake, we conducted a meta-analysis to investigate the impact of supplementation on serum 25-hydroxyvitamin D (25(OH)D) levels in healthy adults in Europe. Of the publications identified (n = 4005) in our literature search (PUBMED, through 2 January 2022), 49 primary studies (7320 subjects, 73 study arms) were eligible for inclusion in our meta-analysis. The risk of bias was assessed using the Cochrane RoB tool based on seven categories, according to which each study is rated using three grades, and overall was rated as rather low. The median duration of intervention was 136.78 days (range, 1088 days); the mean weighted baseline 25(OH)D concentration and mean age were 33.01 vs. 33.84 nmol/L and 46.8 vs. 44.8 years in the vitamin D and placebo groups, respectively. Using random-effects models, 25(OH)D levels were increased by 36.28 nmol/L (95% CI 31.97–40.59) in the vitamin D group compared to the placebo, with a relative serum increment of 1.77 nmol/L per 2.5 μg of vitamin D daily. Notably, the relative serum 25(OH)D increment was affected by various factors, including the dosage and baseline serum 25(OH)D concentration, decreasing with increasing vitamin D doses and with increasing baseline serum levels. We estimate that supplementation in all healthy adults in Europe with appr. 25 μg of vitamin D (1000 IU) daily would raise serum 25(OH)D levels in 95% of the population to ≥50 nmol/L. Our work provides health authorities with reliable data that can help to re-evaluate recommendations for oral vitamin D supplementation.

## 1. Introduction

Vitamin D deficiency affects over one billion children and adults worldwide and is associated with a variety of acute and chronic diseases, including autoimmune diseases, infectious diseases, cardiovascular diseases, cancer, type 2 diabetes, and neurological disorders [1]. Because of its high prevalence and its association with many independent diseases, vitamin D deficiency is now increasingly recognized as a severe health problem [2,3,4]. The human body can acquire vitamin D as vitamin D_2_, derived from irradiated fungal sources, and vitamin D_3_, whose production is induced by ultraviolet B (UV-B) radiation in the skin [2,3,4]. It is now generally accepted that under normal living conditions of populations in temperate parts of the world, most of the population’s vitamin D is synthesized in human skin, while about 20–30% of the population’s vitamin D input comes from the vitamin D that is naturally present in or fortified into food, as well as the intake of dietary supplements [2,3,4]. Vitamin D nutritional status is assessed by measuring the serum 25-hydroxyvitamin D (25(OH)D) concentration. Beyond sun exposure and vitamin D intake, which are the obvious inputs to vitamin D nutritional status, there are a number of lesser contributors to serum 25(OH)D, including underlying diseases; the levels of serum proteins, including vitamin D binding protein; age; skin pigmentation; a culture of minimizing exposure to sunshine; and latitude [5,6]. In northern latitudes from about 40° N, vitamin D production by UV-B rays is minimal during the winter period from October to March when the mid-day solar zenith angle exceeds 45 degrees [6]. In addition, there is also a high prevalence of vitamin D deficiency in Europe in general, independent of the season [7]. Vitamin D nutritional status is lower in Europe than on other continents. But ironically, Scandinavian countries exhibit the highest average serum 25(OH)D levels [3,8,9]. European policies are particularly cautious about fortifying foods with vitamin D and have actually lowered it since the 1950s [6,10].

The aim of this systematic review and meta-analysis is to inform policy makers in Europe about the increases in serum 25(OH)D achieved in healthy European populations in response to the vitamin D doses used in clinical trials.

## 2. Materials and Methods

### 2.1. Search Strategy

This work was conducted according to PRISMA (Preferred Reporting Items for Systematic Reviews and Meta-Analyses) guidelines [11]. We performed an extensive and intentional broad literature search using the online database PUBMED up to 2 January 2022 (Appendix A). MeSH terms and keywords related to vitamin D, vitamin D deficiency, and dietary supplements were used, and the Boolean operators “OR” and “AND” were used to combine these search terms. This search was complemented by a search in the clinical registry clinicaltrials.gov, and reference lists of systematic reviews and meta-analyses for vitamin D supplementation were screened. The screening of titles and abstracts was carried out using the web-based application Rayyan [12]. The full texts of potentially eligible publications were retrieved and screened by hand against the inclusion criteria. To rule out duplicates, all studies were screened by hand.

### 2.2. Inclusion and Exclusion Criteria

Eligible studies were only placebo-controlled primary trials conducted in Europe—RCTs and non-randomized controlled trials (CTs)—with an oral intervention of vitamin D supplementation, regardless of dose, frequency, or whether it was D_2_, D_3_, or 25-hydroxyvitamin D (25(OH)D, or calcidiol). Trials with combined oral supplementation with vitamin D and calcium were also included. All participants in these trials were ≥18 years old and were obviously healthy women or men who lived in Europe. Besides case series, observational studies, systematic reviews and meta-analyses, studies that were not published in English, and studies on animals or cell cultures were excluded. Studies with forms of administration other than oral vitamin D (e.g., intramuscular injection) were excluded, as were studies in which there was no independent vitamin D intervention arm (except for vitamin D given in combination with oral calcium). Studies conducted exclusively with vitamin-D-enriched foods were also excluded (a meta-analysis of food-based fortification with vitamin D has been published elsewhere [13]). Participants <18 years old, pregnant women, or breastfeeding mothers were excluded, as were patients with comorbidities.

### 2.3. Data Extraction and Quality Assessment

The following information was extracted from studies finally included in our meta-analysis: first author; intervention, along with dosage, frequency, and study participants; blinding of the study; country and latitude; age and body mass index (BMI) of participants; ethnicity; duration of the intervention; season in which the study was conducted; baseline and post-intervention serum 25(OH)D concentrations in nmol/L (means and standard deviations (SDs) from each study arm); and the assay method. In the case of missing data, the corresponding authors of the respective studies were contacted and asked to send the missing data. If other units had been reported, such as the median and interquartile range (IQR) or the median and range, the results were converted using the methods of Lou et al. and checked for skewness according to Shi et al. [14,15]. If standard errors (SEs) of the mean or changes from baseline without final serum 25(OH)D concentrations were reported, calculations or estimations were converted into means and SDs according to the recommendations of the Cochrane Collaboration [16]. The Cochrane risk-of-bias tool was used within the Cochrane Review Manager software v.5.4 to assess the risk of systematic bias in the studies [17]. The robustness of the results of the included studies was assessed using the Oxford 2011 Levels of Evidence classification [18].

### 2.4. Statistical Analysis

For the present analyses, a meta-analysis was performed if there were at least 2 studies (arms) in the total population and subgroups. The pre-defined subgroups were analyzed for identification of possible sources of heterogeneity as well as for effect size. Included studies were divided into age categories of 18–59 and ≥60 years. Further subgroups were formed according to baseline serum 25(OH)D concentration threshold values for defining vitamin D deficiency (<50 nmol/L) and sufficiency (≥50 nmol/L), which are widely accepted by professional societies [19]. For a comparison of the different dosing groups, the weighted mean vitamin D dose and serum 25(OH)D concentration were calculated. Moreover, three geographical subgroups were formed according to European latitudes: 36° N to <46° N (southern latitudes), ≥46° N to <55° N (middle latitudes), and ≥55° N to 70° N (northern latitudes). In addition, other subgroups of interest were considered. These were studies with additional calcium supplementation, gender-specific studies, and ethnic minority groups.

In studies with several interventions and/or control arms with partly different doses, respective adjustments were made in accordance with the Cochrane Collaboration recommendations to avoid “unit-of-analysis” errors [16]. For statistical analysis, the inverse variance method and the DerSimonian and Laird random-effects model was chosen [16,20]. The weighted mean difference was expressed in nmol/L, and the weighting of the studies was expressed in percentages. A two-sided statistical significance level was prespecified at α = 0.05. Due to the explorative nature of the study, we report raw *p*-values and did not correct *p*-values or the significance level for the issue of multiple testing. The weighted mean difference and SE were calculated post-intervention from the mean difference in the increase in serum 25(OH)D concentrations in nmol/L plus SD of the intervention group versus the control group, based on the recommendations of the Cochrane Collaboration [16]. The results are presented in both tabular form as well as forest plots. To assess the risk of publication bias, a funnel plot was created and statistically quantified using Egger’s test only if at least 10 studies (arms) were included. Statistical heterogeneity was examined using the Cochran Q test along with the respective two-sided *p*-value based on a chi-square distribution. Heterogeneity was quantified using Higgins and colleagues’ I^2^ statistic [21]. Guidance for assessment is provided by a rough classification into low (I^2^ 0–40%), moderate (I^2^ 30–60%), substantial (I^2^ 50–90%), or considerable (I^2^ > 75%) heterogeneity [16]. Analysis and forest plot generation were performed using Review Manager v.5.4.1 of the Cochrane Collaboration [17]. Funnel plot construction and the calculation of Egger’s test were performed with JASP v.0.16.1 [22]. Additional sensitivity analyses were carried out to evaluate the robustness of the results.

Several formulas were also used or created by the user: (1) the serum increase in nmol/L per 2.5 μg/day vitamin D (α) was calculated using the respective weighted mean (WM) serum 25(OH)D concentration achieved (b), baseline WM serum 25(OH)D concentration (c), and vitamin D dose in μg/day (d) using the following formula from Mo and colleagues [23]:α = [(b − c)/d] × 2.5(1)

(2) To establish a dose recommendation that would result in 95% of the population reaching or maintaining the target serum level, the 5th percentile (β) of the weighted mean baseline serum level of the studies (e) and standard deviation (f) was used with Z-score statistics (with the aid of the standard normal distribution), calculated with the online tool from Soper [24] based on Abramowitz et al. [25], and incorporated into the following formula:β = [e^−(1.644854 × f)^] (2)

(3) The final dose recommendation for vitamin D, which is necessary for 95% of the population to achieve or maintain at least a 50 nmol/L serum 25(OH)D concentration (γ), was calculated using target serum values of 50 nmol/L and 75 nmol/L (g), respectively, the 5th percentile of the WM baseline serum value of the studies using Z-score statistics (β), and the WM serum increase per 2.5 μg/day vitamin D (α) using the following formula:γ = [(g − β)/(α⁄2.5)](3)

## 3. Results

### 3.1. Literature Search and Quality Assessment

The comprehensive literature search (PUBMED, through 2 January 2022) identified a total of 4005 studies and references from systematic reviews and meta-analyses, of which 49 interventional and placebo-controlled studies (48 RCT and 1 CT) that contained a total of 73 vitamin D study arms fulfilled the inclusion/exclusion criteria and could be included in this meta-analysis, as shown in Figure 1. The studies were published from 1995 to 2021, spanning a research period of 26 years. All studies were conducted in Europe.

### 3.2. Study Characteristics

The main characteristics of the 49 studies, with a total of 7320 participants, of whom 3974 received the vitamin D intervention and the remaining 3346 were in placebo arms, are reported in Table 1.

### 3.3. Risk-of-Bias and Quality Assessments

The risk of bias of the 49 included studies in the meta-analysis was assessed and found to be rated as quite low (Figure 2). The Cochrane risk-of-bias tool includes seven categories, for which each study is assessed based on three grades [75]. Only selective reporting was rated as unclear for all studies, as it was not possible to verify whether all prespecified analyses in the protocol were also reported.

The study results showed robustness, and the overall quality of the studies and the resulting evidence levels were high due to the nature of (randomized) interventional trials with fewer limitations and a low risk of bias in the included studies. Missing information in the respective categories was assessed as an unclear risk of bias. A high risk of bias was assigned to the studies concerned, e.g., due to block randomization with an unblinded study design or unblinded personnel, subjects, or data analysis, as well as high dropout rates without more detailed information. According to the 2011 Oxford Level of Evidence Criteria, RCTs are generally rated at level 2, ascending by level of evidence, and CTs are rated at level 3. The level may be downgraded because of study quality, imprecision, indirectness, or inconsistencies between studies or because the absolute effect size is very small; upgrading is possible if there is a large or very large effect size [18]. The 49 included studies were all rated as level 2 or 3 evidence, of which 27 of the studies were rated as level 2 and 22 studies were rated as level 3 using the aforementioned criteria. This also included, in addition to the previously named factors, the assessment of dropout rates (<20% and >20%), which have implications in terms of the consistency of studies and also effect sizes.

Both the evidence assessment and the risk-of-bias assessment on an individual study level are shown in Appendix A.

### 3.4. Meta-Analysis

#### 3.4.1. Characteristics of the Overall Population

The summarized characteristics of the overall study population are balanced across the study arms, as shown in Table 2.

The sample-size-weighted mean values are presented for age at baseline. Weighted and unweighted means are both presented only for the duration of the intervention, which was necessary because the study by Bischoff-Ferrari et al [28]. influenced the duration of the intervention with their large number of subjects and long intervention duration.

#### 3.4.2. Vitamin D Supplementation Increases Serum 25(OH)D Concentration

The increment in 25(OH)D per 2.5 μg/day vitamin D was 1.77 nmol/L. The serum 25(OH)D level significantly increased overall by 36.28 nmol/L (95% CI 31.97, 40.59, *p* < 0.00001) in the vitamin D group compared to the placebo (Figure 3). Heterogeneity was significant (Chi^2^ = 1803.28, *p* < 0.00001) and substantial, with an I^2^ value of 96%, reflecting the wide range of doses used in the included studies. There was no sign of publication bias (Egger’s test *p* = 0.189). The funnel plot is shown in Appendix A.

#### 3.4.3. Subgroup Analysis

Exploratory analyses of the pre-defined subgroups were performed in a multistage manner, comparing different dose categories first and, in a second step, analyzing subgroups within each dose category. This revealed that the nmol/L increase in serum 25(OH)D per unit dose of vitamin D is less in relation to lower age, higher baseline 25(OH)D status, and larger daily vitamin D dose.

1.Vitamin D variants

In an exploratory subgroup analysis, vitamin D supplementation variants D2 and 25(OH)D were examined separately from D3. The weighted mean difference in the serum 25(OH)D increase (36.28 nmol/L (95% CI 31.97, 40.59; *p* < 0.00001)) between vitamin D and the placebo in the total population decreased slightly to 35.78 (95% CI 31.38, 40.18; *p* < 0.00001) after the exclusion of studies with vitamin D2 and 25(OH)D. The subgroups with supplemented vitamin D2 and 25(OH)D in Figure 4 showed a strong increase in serum 25(OH)D compared to the placebo, with a weighted mean difference of 45.75 (95% CI 15.99, 75.50; *p* = 0.003). Despite the fact that only one trial of vitamin D2 was included, it appeared that these results were predominantly due to the effect of the 25(OH)D supplementation trials.

2.Vitamin D supplementation dose

As shown in Table 3, the vitamin D supplementation dose increased, and the relative serum 25(OH)D increment per 2.5 μg supplemented vitamin D decreased. In the population ≥60 years of age, this relative serum 25(OH)D increase was more pronounced (<38 μg/day vitamin D: 5.68; ≥38–<75 μg/day: 1.63; ≥75 μg/day: 0.89 nmol/L) than in the 18–59-year age group (3.31; 1.49; 0.91 nmol/L). This difference was particularly prominent in the low-dose category.

Regardless of the flattening relative serum increment with the increased dose, in the exploratory analyses, especially in the younger population (18–59 years), there was nevertheless a numerical absolute weighted mean difference between vitamin D supplementation and the placebo. This was related to the manifold higher doses in the ≥75 ug/day vitamin D dose category compared to the others.

3.Baseline

The serum 25(OH)D increment was generally more pronounced in subjects aged ≥60 years (<50 nmol/L: 5.77 vs. ≥50 nmol/L: 1.35 nmol/L) than in subjects aged 18–59 years (1.35 vs. 1.07 nmol/L). There was also a consistent picture over all doses with regard to greater serum increments in subgroups with baseline values of <50 nmol/L compared with ≥50 nmol/L (Figure 5).

4.Age

The influence of age on the serum 25(OH)D increment was evident in this work, particularly in the age group ≥60 years. Increasing the dose did not influence the weighted mean difference vs. placebo, although comparisons could only be made between low- and medium-dose categories. In the elderly population, the serum increment per 2.5 μg/day vitamin D was higher than in 18–59-year-olds. This was particularly pronounced in the <38 mcg/day vitamin D dose category, as well as at a baseline 25(OH)D value of <50 nmol/L (Figure 5). Regardless of the gender-specific studies presented in a later chapter, the studies that included only women showed a difference in the serum increment per 2.5 μg/day vitamin D between those ≥60 years of age and those aged 18–59 years. Across all dose categories, as well as within the <38 mcg/day vitamin D dose, there was a significantly greater serum increment per 2.5 μg/day in the group of ≥60-year-old women versus younger women (Table 4). Figure 6 shows the comparison of the two populations—women ≥60 years of age and those aged 18–59 years—at doses <38 mcg/day. The test for subgroup differences suggested that there is a statistically significant subgroup effect (raw *p* = 0.03), meaning that age in women statistically significantly modifies the effect in ≥60-year-olds in comparison to those aged 18–59. The treatment effect favors vitamin D over the placebo for both ≥60-year-old and 18–59-year-old women, although the treatment effect is greater for ≥60-year-old women; therefore, the subgroup effect is quantitative. However, there is considerable heterogeneity between results from the trials within each subgroup that requires further exploration.

5.Ethnicity

The subgroup of ethnic minorities in the northern latitudes is a special subgroup of interest due to an increased risk of severe vitamin D deficiency. Participants were originally from Pakistan, the Middle East, Africa, and South Asia. The subgroups of ethnic minorities living in northern latitudes and the northern population both showed vitamin-D-deficient baseline 25(OH)D levels of <50 nmol/L. However, the ethnic minority subgroup showed severe vitamin D deficiency, with a baseline value of 20.11 nmol/L, compared with a baseline value of 37.60 nmol/L in the northern population, which is almost twice as high, although below 50 nmol/L (Table 5). The serum increment per 2.5 μg/day vitamin D was 3.65 nmol/L in the ethnic minority group. Vitamin D supplementation of 17.43 μg/day resulted in a significant weighted mean difference of 27.62 nmol/L (95% CI 22.35, 32.89; *p* < 0.00001) versus the placebo (Appendix A). Because of the very low baseline level, this barely raised the serum level to the 50 nmol/L threshold.

6.Additional calcium supplementation

The additional calcium supplementation did not show an effect on serum 25(OH)D levels. However, both the age-related higher proportional serum 25(OH)D increment in the elderly compared with the young and a greater increase in those with a baseline <50 nmol/L compared with ≥50 nmol/L were confirmed (Appendix A).

7.Gender

The gender-specific studies in this work were conducted primarily in the 18–59-year-old population only. There were no significant differences between women and men in terms of the serum increment per 2.5 μg/day vitamin D in the different dose categories. However, again, the strongest serum increment per 2.5 μg/day was seen in the vitamin D dose category <38 μg/day and decreased with increasing doses (Table 6).

#### 3.4.4. Sensitivity Analysis

To check the robustness of the results and to further identify possible causes of heterogeneity, a sensitivity analysis was performed. In this analysis, each individual study in the entire study (arm) group was removed one after the other to check whether there was an influence. This procedure, however, did not reveal any discrepancies in the results. Furthermore, studies with large numbers of subjects ≥100 [28,47,49,57,60,64] were excluded to examine the effect of these large studies on the overall collective. Similarly, this approach did not reveal a meaningful difference in the results. Going further, we then used the established inclusion criteria to examine whether excluding studies in which calcium was also administered [26,29,30,34,41,64,68] had an effect on the results. This exclusion did not change the results either. In addition to this, the study arms whose interventions did not include vitamin D3 were then also excluded [33,43,67]. Neither their joint exclusion with the calcium studies nor their exclusion alone changed the results.

#### 3.4.5. Estimation of the Vitamin D Supplementation Dose to Achieve the Desired Serum 25(OH)D Concentration

Figure 7 shows, for all included studies, the scatter plot of the final serum 25(OH)D values and their respective daily doses of vitamin D. Rising final serum 25(OH)D levels are shown with increasing daily vitamin D dosing. A total of 38% of the studies were able to achieve final mean serum 25(OH)D levels of at least 50 nmol/L with the vitamin D dosage specified in these studies. The study by Wyon et al. (2016) [72] was identified as an outlier and was excluded from t”e gr’ph. The reason for this was the single bolus of vitamin D of 3750 μg with a 7-day observation, which resulted in the herein-calculated dose of 535.7 μg of vitamin D. On the other hand, the 25(OH)D values shown as means minus 2 SD were calculated as negative values for the supplemented arms of the studies by Wyon et al. (2021, “Liquid” arm) [73], and the study by Zittermann [74] actually showed a reduction in final serum 25(OH)D levels. The most likely cause of the negative values for mean minus 2 SD is poor compliance among study subjects, as implied by the fact that the standard deviation was dramatically increased in the vitamin D groups compared to baseline values and the placebo group’s final values. Despite the compliance issue, an additional explanation for the “Liquid” arm of the study from Wyon et al. was seen by the authors in the application of 100.000 IU vitamin D over a 24 h period, whereas the reduced efficacy of a bolus of oral liquid versus a slower-release pill may be due to the rate-limited hepatic hydroxylation of vit D to 25(OH)D following rapid intestinal absorption. In particular, if taken all at once, this may have saturated the absorption capacity of the intestine, so a dosed intake of three intakes over a 24 h period might have been more effective [73]. Other points of interest in Figure 7 appear in the two strongest post-intervention serum 25(OH)D increments at daily doses up to 50 mcg/day vitamin D (the two triangles within the scatter plot). On the one hand, this applied to the arm of the study by Cashman et al. with 20 μg/day 25(OH)D, whose 4-5-fold higher potency compared with vitamin D3 was confirmed here once again [33], and on the other hand, in the study arm of the older population in the study by Agergaard et al., a supplemented daily dose of vitamin D of 48 μg showed a further significant increase in serum 25(OH)D levels, even at a baseline value as high as 70 nmol/L [26].

Calculations with the above formulas resulted in an estimated dose of 24.90 ug/day vitamin D for the total population, a higher dose of 35.91 μg for the younger group, and a lower dose of 15.49 μg for the older group (Table 7).

## 4. Discussion

Because of its high prevalence and its association with many independent diseases, vitamin D deficiency represents a major obstacle to human health worldwide. To generate or re-evaluate recommendations for the oral uptake of vitamin D, health authorities in Europe and other countries at present have an urgent need to obtain reliable data on the effect of vitamin D supplementation on vitamin D status. It is generally accepted that the impact of vitamin D supplementation depends on various regional factors, which include ethnicity, geographical region, and season. Because we were particularly interested in analyzing the situation in the European Union, we performed a literature search (PubMed), meta-analysis, and systematic review restricted to investigating the effect of supplementation on serum 25-hydroxyvitamin D (25(OH)D) levels in healthy adults in Europe. Assessing the dose-related weighted mean difference and standard error using random-effects models, we found that serum 25(OH)D levels increased significantly by 36.28 nmol/L (95% CI 31.97, 40.59, *p* < 0.00001) in the vitamin D group compared to the placebo in the 49 studies included, with a relative serum increment of 1.77 nmol/L per 2.5 μg of vitamin D daily. Notably, the relative serum 25(OH)D increment was affected by various factors, including dosage and baseline serum 25(OH)D concentration, decreasing with increasing vitamin D doses and with increasing baseline serum levels. We estimate that vitamin D supplementation at appr. 25 μg (1000 IU) daily in all adults in Europe would raise serum 25(OH)D levels in 95% of the population to ≥50 nmol/L. Our work contributes to the increasing body of evidence that provides health authorities with reliable data that allow them to re-evaluate recommendations for oral vitamin D supplementation.

Vitamin D deficiency is prevalent globally, including in Europe. Population studies showed a vitamin D status of <50 nmol/L in 24% of the United States [76], 36.8% of Canada [77], and 40.4% of Europe [7]. Especially at northern latitudes from about 40° N, sunlight is not strong enough to trigger the synthesis of vitamin D in the skin from October to March [6]. Within Europe, serum 25(OH)D levels are higher in northern Europe, which may be related to traditionally higher consumption of fatty fish and cod liver oil, whereas low serum 25(OH)D levels in southern European countries may be due to greater skin pigmentation and sunshine-avoiding behavior [78]. We agree with Frost [79] that metabolic processes are affected by natural selection. In consequence, humans living at higher latitudes may have adapted to reduced synthesis of vitamin D. These adaptions may include higher uptake of calcium from the intestines, a higher rate of conversion of vitamin D to its active forms, stronger binding of vitamin D to carrier proteins in the bloodstream, and greater use of alternative metabolic pathways for calcium intake. However, we also agree with others [80] that human health is affected by vitamin D deficiency if the serum 25(OH)D level is below 20 ng/mL. To control micronutrient malnutrition, the WHO-FAO (World Health Organization and Food and Agriculture Organization) has suggested a greater variety of micronutrient-containing foods, increased food fortification, and supplementation as strategies [81]. With respect to vitamin D, both the quantity and range of diverse food sources are challenging because very few contain sufficient vitamin D [82]. The WHO-FAO suggests that supplementation often provides the most rapid improvement in micronutrient status for individuals or target populations, but food fortification usually has a less immediate yet much more comprehensive and lasting effect [81]. The aim of the present systematic review and meta-analysis was to investigate the effect of oral vitamin D supplementation on serum 25(OH)D levels in healthy adult subjects in Europe and also to derive from this a dose recommendation for specific populations to achieve or maintain adequate vitamin D levels.

### 4.1. Influencing Factors on the Serum 25(OH)D Increment

The magnitude of the serum 25(OH)D increment in this work was dependent on factors including age, dose, and baseline 25(OH)D level. The vitamin D dose–response has a curvilinear rather than a linear serum 25(OH)D increment that, above a certain dose, causes serum concentrations to no longer increase linearly [83,84]. It was shown that baseline 25(OH)D levels ≥50 nmol/L require more vitamin D for a serum increment than those with a <50 nmol/L baseline serum 25(OH)D concentration [83]. A global meta-analysis that also included patients with comorbidities showed comparable dose–effect results, with relative serum increments decreasing with increasing doses, and showed that older populations and people with lower baseline 25(OH)D levels had greater serum increments [23].

The results for the entire subgroup of 18–59-year-olds showed a serum increment of 1.26 nmol/L per 2.5 μg of vitamin D. In the study by Heaney and colleagues, this translated to 1.75 nmol/L per 2.5 μg of vitamin D (0.7 nmol/L per μg) needed to maintain, achieve, or both achieve and maintain serum levels of 80 nmol/L in winter in a collective of males aged 38.7 years ± SD 11.2 from Omaha, Nebraska, USA, at latitude 41.2° N [85].

The results in the population aged ≥60 years also confirm those from Mo and colleagues as well as McKenna and Murray. Both found a greater serum increment in the older subgroup compared with younger adults [23,86]. Mo et al. also saw a greater increase with lower doses within age groups [23]. Furthermore, Whiting and colleagues described a serum 25(OH)D increment of 5.53 nmol/L per 2.5 μg/day vitamin D in their systematic review of included studies conducted predominantly with older populations and low doses (5 to 20 μg/day vitamin D) [87]. In our work, the relative serum increment per 2.5 μg of vitamin D for the subgroup of ≥60-year-olds with low doses was 5.68 nmol/L and was in line with the findings of Whiting and colleagues. In addition, a study by Barger-Lux and colleagues also showed a greater serum increment in the older versus younger population [88]. This is also underpinned by our results, demonstrating that age is an influencing factor on the serum 25(OH)D increment.

A subgroup analysis of ethnic minorities from northern latitudes revealed the particular risk of vitamin D deficiency in these populations. The mean baseline 25(OH)D level of 20.11 nmol/L was in the range of severe vitamin D deficiency. Although the relative serum increment was 3.65 nmol/L per 2.5 μg/day vitamin D, the mean vitamin D dose of 17.43 μg/day was not sufficient to raise the target serum level to ≥50 nmol/L. Non-Western immigrants are in the at-risk group for vitamin D deficiency [19]. The risk factors for vitamin D deficiency in non-Western immigrants in more northern latitudes mainly include lower UV-B radiation exposure than in their home country, dark skin pigmentation, latitude before emigration, length of stay in the hosting country, cultural habits such as wearing a veil, different dietary habits, low intake of vitamin D supplements, and low calcium intake [89]. To raise the serum 25(OH)D concentration in 95% of this population to at least 50 nmol/L, our dose recommendation requires vitamin D supplementation of 19.17 μg/day.

The results of this meta-analysis showed no significant effects of additional calcium supplementation on the absolute serum 25(OH)D increment. This is consistent with the results of the work by Mo and colleagues [23]. The consideration of calcium in combination with vitamin D within the scope of this work referred exclusively to the serum 25(OH)D increment; the additive effects of calcium are not in the scope of this work.

In our gender-specific comparison of vitamin D dose categories, there was, in general, no relevant difference in the relative serum increment between women and men. However, it was observed in our work that the baseline 25(OH)D level tended to be lower in females compared with males. Nevertheless, a comparison was only feasible in the age group of 18- to 59-year-olds. There are discrepancies in the literature regarding gender differences in serum 25(OH)D levels. The large Euronut Seneca study investigated 12 European countries and found that older women have a greater proportion of severe vitamin D deficiency (<30 nmol/L): 47% compared with 36% in men [90]. This was also confirmed in the National Diet and Nutrition Survey in the United Kingdom, where even younger women were more affected by severe vitamin D deficiency (<25 nmol/L) than men. In the case of vitamin D deficiency (<50 nmol/), the higher proportion of women compared with men was only evident from >50 years of age [91]. In contrast, Cashman and colleagues’ reanalysis of 14 European population studies in the International Vitamin D Standardization Program showed no relevant differences in the range of severe vitamin D deficiency (<30 nmol/L) in the overall population between women and men, either younger or older [7].

### 4.2. Estimation of Vitamin D Supplementation Dosage

Our vitamin D dose recommendation to ensure that 95% of the population reaches or maintains a ≥50 nmol/L 25(OH)D serum level is 24.90 ug/day for the overall population, with a higher dose recommendation for the younger population (35.91 ug/day) and a lower one for the older population (15.49 ug/day). The dose recommendations are in line with those of Mo and colleagues [23] when based on the target value of 75 nmol/L, as shown in Table 7.

Our dose recommendation for those ≥60 years of age is slightly below the recommendations of European countries in the DACH region (Germany, Austria, and Switzerland), the Nordic countries (including Denmark, Finland, Sweden, and Norway), the Netherlands, and Spain and in the upper range of recommendations, ranging from 5 to 15 μg/day, from France, Belgium, the United Kingdom, and Ireland, as well as EFSA, the European Food Safety Authority [19]. These guidelines listed by Lips et al. mostly use a serum 25(OH)D target of at least 50 nmol/L. The ECTS (European Calcified Tissue Society) itself recommends 10–20 μg/day vitamin D for those >70 years of age [19]. The Institute of Medicine (IOM) in the USA recommends 20 μg/day vitamin D for “elders”, defined as >70 years [84]. The American Endocrine Society does recommend 15–20 μg/day vitamin D for both adults aged 50–70 and those >70 years. However, 37.5 to 50 μg/day vitamin D is explicitly recommended to raise serum 25(OH)D levels to >75 nmol/L [80].

Within the 18–59-year-old group, our dose recommendation deviated more markedly from the previously mentioned European guidelines. Also, compared with the Endocrine Society’s dose recommendation of 37.5–50 μg/day vitamin D, with the target serum level of 75 nmol/L, the dose recommended here to reach or maintain a 75 nmol/L serum 25(OH)D concentration is 64.39 μg/day (~2575.69 IU/day) vitamin D above this recommendation. In a meta-analysis of individual patient data by Cashman and colleagues, they showed that the calculated and recommended daily vitamin D doses were 2.5-fold higher than the recommendations of the IOM, Nordic nutrition recommendations (NORDEN), and EFSA panels in some cases to ensure that 97.5% of the population aged 4–90 years can reach or maintain at least a 50 nmol/L serum 25(OH)D concentration [92]. In the mentioned guidelines of European countries, as well as those of professional societies, vitamin D doses for adults are equivalent to or 5–10 μg/day vitamin D lower than the recommendations for the older population [19]. The Endocrine Society considers the 19–50 age group to be at risk of vitamin D deficiency, since reduced time spent outdoors and extensive sun protection are additional factors [80]. In addition, vitamin D deficiency often remains undetected in adults aged 18–59 years due to a lack of regular serum level testing. Consequently, possible vitamin D secondary diseases or already immanent diseases remain hidden. Moreover, since in Europe, food is hardly fortified with vitamin D, vitamin D deficiency is more prevalent in Europe than on other continents.

### 4.3. Vitamin D Intoxication

The topic of toxicity should generally also be considered in the area of vitamin D. A vitamin D dose should be targeted that can improve vitamin D status while avoiding or minimizing the risks of potential toxicity associated with overdose [93]. Vitamin D intoxication is associated with hypercalcemia, hyperphosphatemia, and suppressed PTH levels, which typically occurs with the excessive consumption (1250 μg/day to 25,000 μg/day) of vitamin D over several months to years [10]. Borderline serum 25(OH)D levels that can cause vitamin D intoxication are seen at 375 nmol/L [94,95]. But even when hypercalcemia is detected, it needs to be kept in mind that the upper reference value for calcium is defined as a level above the 97.5th percentile of the apparently healthy, local reference population [96]. Vitamin D intoxication is therefore described as very rare [6,94,97]. However, the upper limits of vitamin D intake from supplements are very diverse in the recommendations of different bodies. While the Endocrine Society defines this for adults at 250 μg/day [80], EFSA and IOM see it as low as 100 μg/day [84,98], and the European Union uniformly defines it at 50 μg/day vitamin D from 11 years of age [6]. According to the IOM recommendations, 100 μg/day vitamin D does not lead to mean serum 25(OH)D concentrations above 125 nmol/L [84]. According to Vieth, the low upper limits of some recommendation committees are historically based [99], and that precedent represents a psychological barrier to change in successor committees, so the low upper limit itself may be the problem, as committees only increase the uncertainty factor, not the upper limit, in evaluating safe, higher doses [100]. In the context of this meta-analysis, only the studies by Kjaergaard [47], Kujach [50], Grimnes [39], and Prietl [63] exceeded this serum level of IOM, but there was no evidence of side effects suggestive of vitamin D intoxication in any of the studies.

### 4.4. Strengths and Limitations

A strength of this systematic review and meta-analysis of oral vitamin D supplementation is its focus on healthy adults in Europe and its effect on serum 25(OH)D levels. From this, dose recommendations for vitamin D supplementation were calculated in order to address vitamin D deficiency in the European population. These should be suitable for increasing 25(OH)D to optimal serum concentration ranges. In this way, this work also aims to provide suggestions for the prevention of vitamin-D-deficiency-associated secondary diseases, including through adequate vitamin D supplementation.

Nevertheless, this work also has limitations. One of these is the overall high heterogeneity of the study results. It was not possible to exclude further possible confounding factors due to a lack of information in the studies. In connection with vitamin D, these include, above all, skin type, diet, additional vitamin D intake as such, and also the influence of sunlight on the subjects. Unfortunately, it was not possible to obtain more detailed information to consider these characteristics in this work. In addition, the heterogeneity may also relate to the different characteristics of the participants, the duration and frequency of vitamin D intake, and the vitamin D dose, as well as the respective 25(OH)D measurement method. Various methods for 25(OH)D concentration analysis were used in the studies, including CBPA (competitive protein-binding assay), CLIA (chemiluminescence immunoassay), ECLIA (electrochemiluminescence immunoassay), ELISA (enzyme-linked immunosorbent assay), RIA (radioimmunoassay), and LC-MS/MS (liquid chromatography–mass spectrometry/mass spectrometry), although results were validated by proficiency testing. In addition, no specific time of year was specified in the present work, which of course has an influence on serum 25(OH)D levels in addition to the above factors.

A limitation may also be the duration of vitamin D supplementation. There were no restrictions in the inclusion and exclusion criteria of this study, so studies with a single dose were included, as well as studies with a supplementation duration of one year or longer. A distortion, including in an interaction with the previously mentioned factors, including the time of year, can therefore not be completely ruled out.

Furthermore, another limitation of this work was that the literature search and selection of studies were conducted by only one person. Since a control and a comparison for the literature search and the study inclusion by another person did not take place, the possibility of a distortion exists, as well as a possibility that studies were missed.

## 5. Conclusions

This systemic review and meta-analysis assembled and analyzed data available specifically for Europe, facilitating the development of recommendations for vitamin D and calcidiol intakes for European populations. Based on the evidence presented here, we conclude that, to ensure serum 25(OH)D levels of at least 50 nmol/L in 95% of the population, all healthy adults in Europe would need an additional uptake of at least 25 μg of vitamin D3 (1000 IU) daily, which could be provided by supplementation and/or food fortification. However, further studies, e.g., in ethnic minorities, are needed to ensure that specific subgroups receive adequate amounts of vitamin D. Other results of this study that are in agreement with previous work include findings that the effect of vitamin D supplementation on serum 25(OH)D concentration depends on various independent factors, including age, the dose of vitamin D supplementation, and the baseline 25(OH)D concentration. We conclude that further research on this topic, including our finding of a decrease in the relative serum 25(OH)D increment per 2.5 ug supplemented vitamin D with increasing doses of vitamin D (Table 3), is needed. It should be noted that in this systematic review and meta-analysis, we were unable to find any evidence that the higher doses of vitamin D supplementation used were associated with a higher frequency or severity of adverse events. Considering the safety, easy availability, low costs, and potentially enormous positive health effects, this study adds to the continuously growing body of evidence that should provide health authorities with arguments to recommend an additional uptake of at least 25 μg of vitamin D3 (1000 IU) daily in the adult European population, which could be provided by supplementation and/or food fortification.

## Figures and Tables

**Figure 1 nutrients-15-03986-f001:**
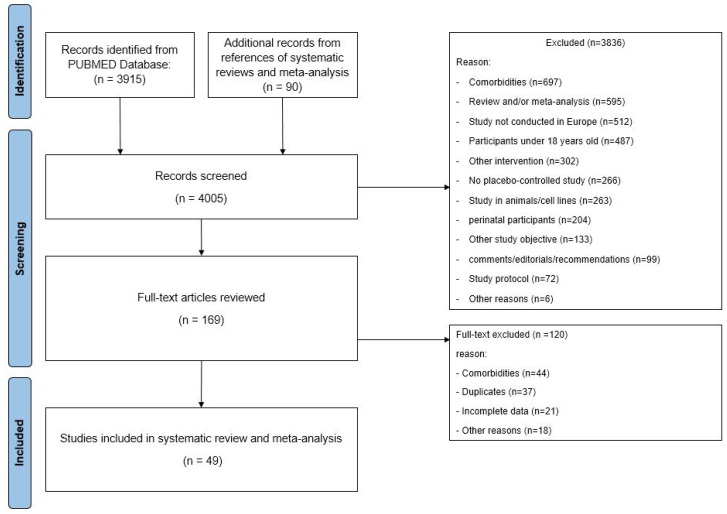
Flow chart according to PRISMA guidelines [11] showing the structured process, from PUBMED search results to title and abstract screening against inclusion/exclusion criteria to full-text reading of prior positively screened studies for further alignment with inclusion/exclusion criteria for the final inclusion of studies for systematic review and meta-analysis.

**Figure 2 nutrients-15-03986-f002:**
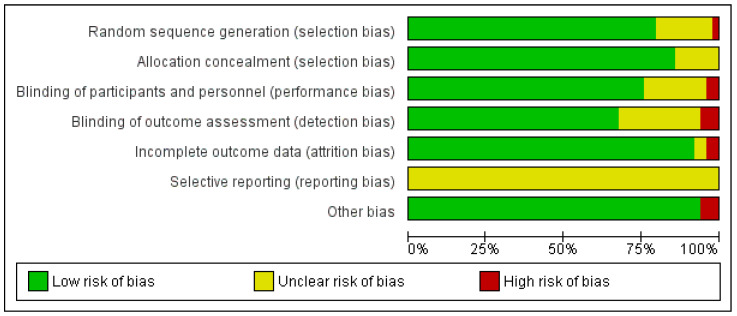
Risk-of-bias assessment—overall summary.

**Figure 3 nutrients-15-03986-f003:**
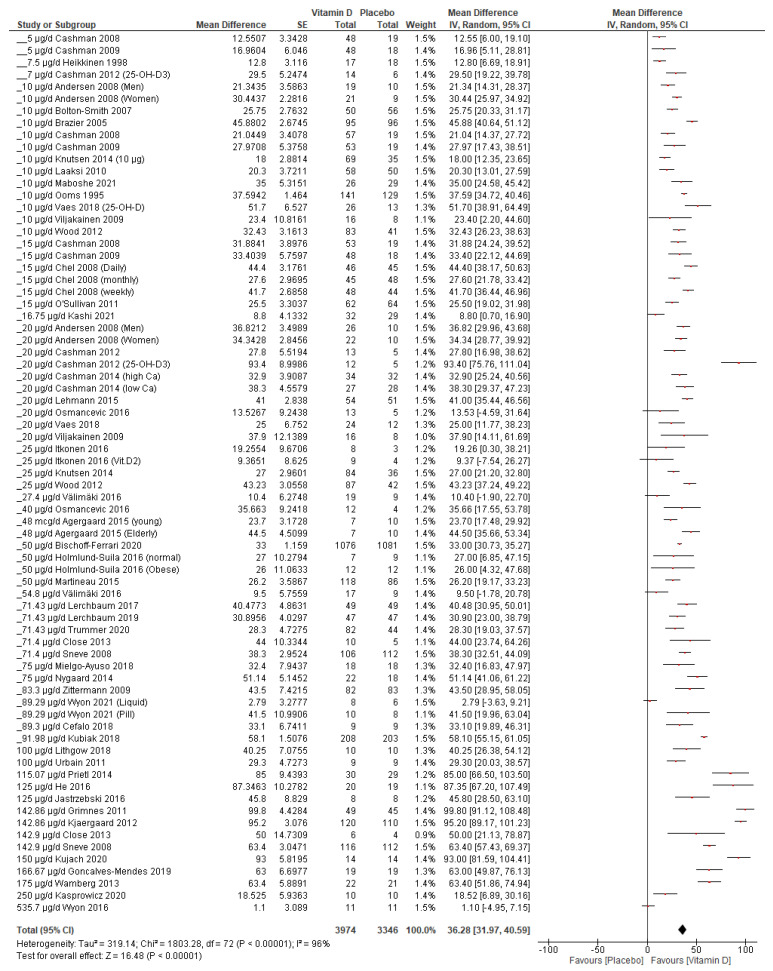
Forest plot of all included studies (arms). Created with Review Manager v.5.4.1 [17]. The effect of vitamin D supplementation (right side of the graph) on serum 25(OH)D increment compared with placebo was demonstrated in all studies [26,27,28,29,30,31,32,33,34,35,36,37,38,39,40,41,42,43,44,45,46,47,48,49,50,51,52,53,54,55,56,57,58,59,60,61,62,63,64,65,66,67,68,69,70,71,72,73,74]. With the exception of 6 studies that crossed the zero line, this effect was also statistically significant [43,61,68,72,73].

**Figure 4 nutrients-15-03986-f004:**
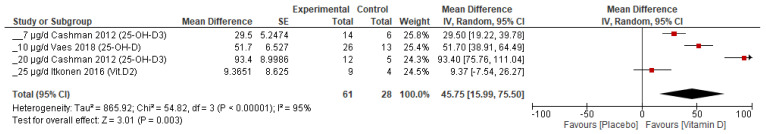
Subgroups of vitamin D variants 25(OH)D and D2. The effect of serum 25(OH)D increment vs. placebo was stronger in the studies with 25(OH)D supplements than in those with D2. The vitamin D2 study, furthermore, could not show a statistically significant serum increment over placebo due to crossing the zero line [33,43,67].

**Figure 5 nutrients-15-03986-f005:**
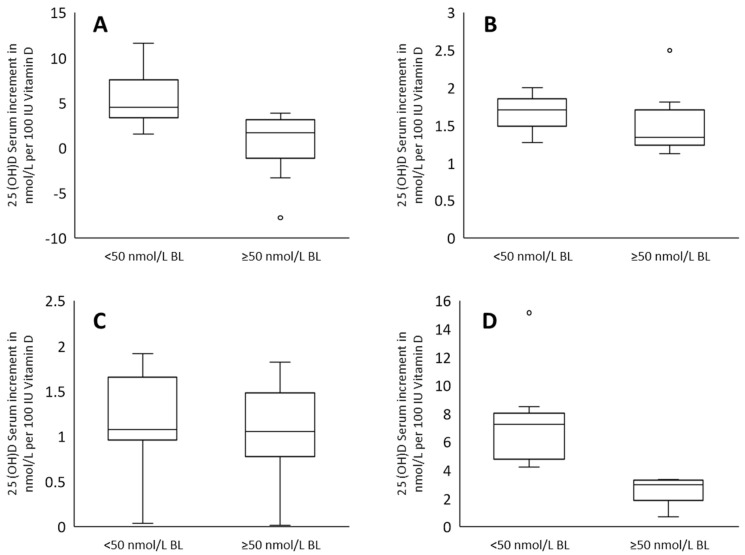
Within each box, horizontal lines denote median values; boxes extend from the 25th to the 75th percentile of each group’s distribution of values; vertically extending lines denote adjacent values (i.e., the most extreme values within 1.5 interquartile range of the 25th and 75th percentiles of each group); dots denote observations outside the range of adjacent values. The relative serum 25(OH)D increment per 2.5 μg daily vitamin D dependent on baseline 25(OH)D levels <50 and ≥50 nmol/L are shown for (**A**) age 18–59 years with <38 μg/day vitamin D, (**B**) age 18–59 years with ≥38–<75 μg/day vitamin D, (**C**) age 18–59 years with ≥75 μg/day vitamin D, and (**D**) age ≥60 years with <38 μg/day vitamin D. On the one hand, it was shown that baseline 25(OH)D levels <50 nmol/L achieved a greater serum increment per 2.5 μg of vitamin D (100 IU) per day regardless of the supplemented vitamin D dose. On the other hand, however, it also appeared that this relative serum increment was most pronounced in the low-dose category and always decreased with higher doses. In addition, it was also shown that the older population achieved a greater serum increment per 2.5 mcg of daily vitamin D compared to the younger population with low-dose vitamin D.

**Figure 6 nutrients-15-03986-f006:**
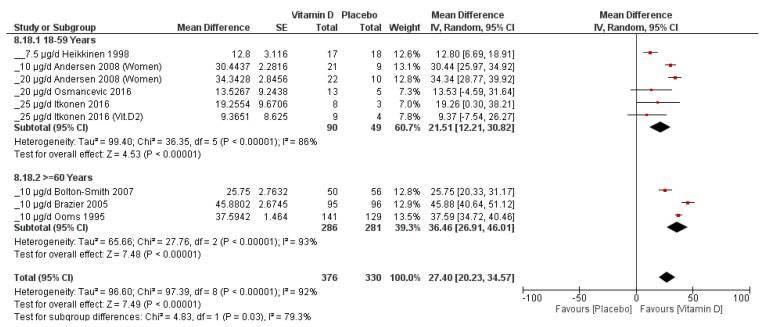
Subgroup analysis of women with low dose, 18–59 vs. ≥60 years. Created with Review Manager v.5.4.1 [17]. Both subgroups showed significant weighted mean difference between vitamin D supplementation versus placebo (both *p* < 0.00001), despite significant and substantial heterogeneity. The serum increase with vitamin D supplementation was significantly greater in the older population than in the younger population (*p* = 0.03), although the heterogeneity in the results of the studies of both subgroups could not be explained by this subgroup analysis [27,29,30,41,43,60,61].

**Figure 7 nutrients-15-03986-f007:**
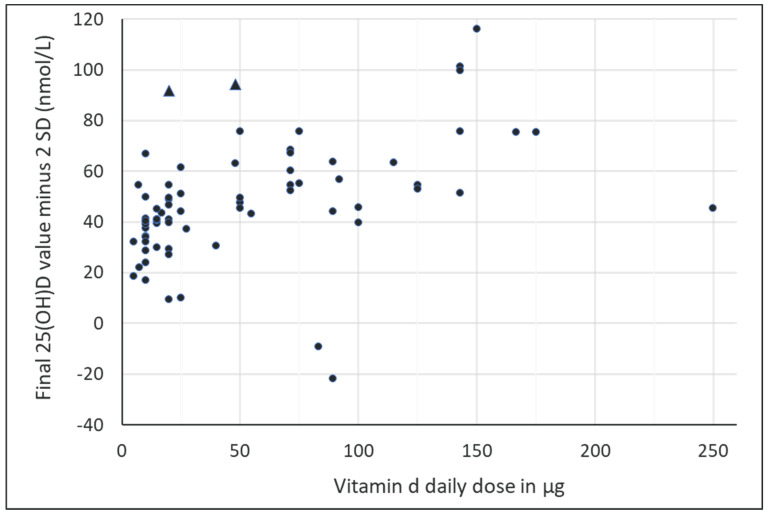
Estimation of serum 25(OH)D concentration reasonably assured for adults taking the doses used in European clinical trials of vitamin D supplementation using cholecalciferol (circles) or calcidiol (triangles). The data points shown in this scatter plot each show the values of the mean minus 2 SD for serum 25(OH)D in all vitamin D studies included in this meta-analysis versus the daily doses of vitamin D in those studies. That is, each data point represents the lowest serum 25(OH)D value estimated for the study’s daily dose.

**Table 1 nutrients-15-03986-t001:** Main characteristics of included studies.

First Author and Year	Type of Vitamin D	Average Dose μg/Day	Blinding	Country	Degrees North Latitude	Age Category	Duration of Intervention	Season	ITT (N)	Oxford Level of Evidence
Agergaard 2015 [26]	D3	48	Double-blind	Denmark	≥55° N to 70° N	18–59 and ≥60	16 Weeks	Nov–Apr	34	2
Andersen 2008 [27]	D3	10/20	Double-blind	Denmark	≥55° N to 70° N	18–59	52 Weeks	Jan–Nov	173	3
Bischoff-Ferrari 2020 [28]	D3	50	Double-blind	Switzerland, France, Portugal, Germany, Austria	≥46° N to <55° N	≥60	156 Weeks	n.a.	2157	2
Bolton-Smith 2007 [29]	D3	10	Double-blind	Great Britain	≥55° N to 70° N	≥60	104 Weeks	n.a.	123	2
Brazier 2005 [30]	D3	10	Double-blind	France	36° N to <46° N	≥60	52 Weeks	n.a.	192	3
Cashman 2008 [31]	D3	5/10/15	Double-blind	Ireland/Great Britain	≥46° N to <55° N	18–59	22 Weeks	Oct–Apr	245	2
Cashman 2009 [32]	D3	5/10/15	Double-blind	Ireland	≥55° N to 70° N	≥60	22 Weeks	Oct–Mar	216	2
Cashman 2012 [33]	25(OH)D3/D3	7/20/20	Double-blind	Ireland	≥46° N to <55° N	18–59	10 Weeks	Jan–Apr	58	2
Cashman 2014 [34]	D3	20	Double-blind	Ireland	≥46° N to <55° N	18–59	15 Weeks	Nov–Mar	125	2
Cefalo 2018 [35]	D3	89.3	Double-blind	Italy	36° N to <46° N	18–59	13 Weeks	n.a.	18	3
Chel 2008 [36]	D3	15	Open	Netherlands	≥46° N to <55° N	≥60	17 Weeks	n.a.	338	3
Close 2013 [37]	D3	71.4/142.9	Double-blind	Great Britain	≥46° N to <55° N	18–59	12 Weeks	Jan–Apr	30	3
Goncalves-Mendes 2019 [38]	D3	166.67	Double-blind	France	≥46° N to <55° N	≥60	13 Months	Jun–Oct	40	3
Grimnes 2011 [39]	D3	142.86	Double-blind	Norway	≥55° N to 70° N	18–59	26 Weeks	n.a.	104	2
He 2016 [40]	D3	125	Double-blind	Great Britain	≥46° N to <55° N	18–59	14 Weeks	n.a.	39	3
Heikkinen 1998 [41]	D3	7.5	Open	Norway	≥55° N to 70° N	18–59	52 Weeks	n.a.	35	3
Holmlund-Suila 2016 [42]	D3	50	Double-blind	Finland	≥55° N to 70° N	18–59	12 Weeks	Nov–May	42	3
Itkonen 2016 [43]	D2/D3	25/25	Double-blind	Finland	≥55° N to 70° N	18–59	8 Weeks	Feb–Apr	31	3
Jastrzebski 2016 [44]	D3	125	Double-blind	Poland	≥46° N to <55° N	18–59	4 Weeks	Mar–Apr	16	3
Kashi 2021 [45]	D3	16.75	Double-blind	Great Britain	≥55° N to 70° N	18–59	12 Weeks	n.a.	61	3
Kasprowicz 2020 [46]	D3	250	Double-blind	Poland	≥46° N to <55° N	18–59	2 Weeks	Autumn	20	3
Kjaergaard 2012 [47]	D3	142.86	Double-blind	Norway	≥55° N to 70° N	18–59	26 Weeks	n.a.	230	2
Knutsen 2014 [48]	D3	10/25	Double-blind	Norway	≥55° N to 70° N	18–59	16 Weeks	Jan–Jun	251	2
Kubiak 2018 [49]	D3	91.98	Double-blind	Norway	≥55° N to 70° N	18–59	17 Weeks	n.a.	411	2
Kujach 2020 * [50]	D3	150	Single-blinded	Polen	≥46° N to <55° N	18–59	8 Weeks	Jan–Mar	28	3
Laaksi 2010 [51]	D3	10	Double-blind	Finland	≥55° N to 70° N	18–59	26 Weeks	Oct–Mar	164	3
Lehmann 2015 [52]	D3	20	Double-blind	Germany	≥46° N to <55° N	18–59	12 Weeks	Jan–Apr	106	2
Lerchbaum 2017 [53]	D3	71.43	Double-blind	Austria	≥46° N to <55° N	18–59	12 Weeks	n.a.	100	2
Lerchbaum 2019 [54]	D3	71.43	Double-blind	Austria	≥46° N to <55° N	18–59	12 Weeks	n.a.	100	2
Lithgow 2018 [55]	D3	100	Double-blind	Great Britain	≥55° N to 70° N	18–59	6 Weeks	Oct–Jun	20	2
Maboshe 2021 [56]	D3	10	Double-blind	Great Britain	≥55° N to 70° N	18–59	43 Weeks	Mar–Jan	59	2
Martineau 2015 [57]	D3	50	Double-blind	Great Britain	≥46° N to <55° N	≥60	52 Weeks	n.a.	240	2
Mielgo-Ayuso 2018 [58]	D3	75	Double-blind	Spain	36° N to <46° N	18–59	8 Weeks	Apr–Jun	36	2
Nygaard 2014 [59]	D3	75	Double-blind	Denmark	≥55° N to 70° N	18–59	16 Weeks	Dez–Apr	50	2
Ooms 1995 [60]	D3	10	Double-blind	Netherlands	≥46° N to <55° N	≥60	52 Weeks	n.a.	348	3
Osmancevic 2016 [61]	D3	20/40	Double-blind	Sweden	≥55° N to 70° N	18–59	12 Weeks	n.a.	114	3
O’Sullivan 2011 [62]	D3	15	Double-blind	Ireland	≥46° N to <55° N	18–59	4 Weeks	n.a.	126	2
Prietl 2014 [63]	D3	115.07	Double-blind	Austria	≥46° N to <55° N	18–59	13 Weeks	n.a.	60	2
Sneve 2008 [64]	D3	71.4/142.9	Double-blind	Norway	≥55° N to 70° N	18–59	52 Weeks	n.a.	445	3
Trummer 2020 [65]	D3	71.43	Double-blind	Austria	≥46° N to <55° N	18–59	24 Weeks	n.a.	150	2
Urbain 2011 [66]	D3	100	Single-blinded	Germany	≥46° N to <55° N	18–59	5 Weeks	Jan–Mar	18	3
Vaes 2018 [67]	25(OH)D3/D3	10/20	Double-blind	Netherlands	≥46° N to <55° N	≥60	26 Weeks	Dez–Dez	78	2
Välimäki 2016 [68]	D3	27.4/54.8	Open	Finland	≥55° N to 70° N	≥60	52 Weeks	Mar–Mar	60	3
Viljakainen 2009 [69]	D3	10/20	Double-blind	Finland	≥55° N to 70° N	18–59	26 Weeks	Nov–Apr	48	3
Wamberg 2013 [70]	D3	175	Double-blind	Denmark	≥55° N to 70° N	18–59	26 Weeks	n.a.	52	3
Wood 2012 [71]	D3	10/25	Double-blind	Great Britain	≥55° N to 70° N	≥60	52 Weeks	Jan–Jan	305	2
Wyon 2016 [72]	D3	535.7	Double-blind	Great Britain	≥46° N to <55° N	18–59	1 Week	Jan	22	2
Wyon 2021 [73]	D3	89.29	Double-blind	Great Britain	≥46° N to <55° N	18–59	4 Weeks	Mar	40	2
Zittermann 2009 [74]	D3	83.3	Double-blind	Germany	≥46° N to <55° N	18–59	52 Weeks	n.a.	200	2

*: Non-randomized trial. Abbreviations: CT = (non-randomized) controlled study; ITT = intention-to-treat population; n.a. = not available; RCT = randomized controlled trial.

**Table 2 nutrients-15-03986-t002:** Characteristics of the overall study population.

Baseline Characteristics of Subjects from Overall Study Population (49 Studies with 73 Study Arms)		
	Vitamin D	Placebo
Participants, N	3974	3346
Mean age (range) in years	59.2 (20–84)	59.6 (20–84)
Percentage women vs. men	61 vs. 39%	
Mean duration of intervention (range) in days Unweighted mean duration of intervention (range) in days	452.4 (7–1095) 183.9 (7–1095)	
Mean baseline 25(OH)D (range) in nmol/L	47.04 (10–79)	47.4 (13–80)
Mean daily dose (range) in μg	51.8 (5–536)	51.8 (5–536)

All values are weighted by study size if not otherwise specified.

**Table 3 nutrients-15-03986-t003:** Characteristics of study populations by age and dose category/baseline 25(OH)D.

Age in Years	Dose Category	Baseline 25(OH)D Category	Study Arms, N	Dose, μg/Day *	Vitamin D Group, N	Baseline 25(OH)D, nmol/L *	Weighted Mean Difference in nmol/L (95% CI)	Serum Increment in nmol/L per 2.5 μg/Day Vit.D. ^ƚ^	*p*-Value of Serum Increment
18–59	<38 μg/day (A)		25	15.23	810	43.79	27.53 (22.99, 32.07)	3.31	A vs. B *p* = 0.037
	≥38–<75 μg/day (B)		9	63.21	332	50.75	32.44 (27.29, 37.58)	1.49	B vs. C *p* = 0.039
	≥75 μg/day (C)		20	142.90	782	43.57	51.63 (37.53, 65.74)	0.91	A vs. C *p* = 0.001
		<50 nmol/L	31	75.86	1035	33.05	39.00 (29.84, 48.17)	1.35	*p* = 0.035
		≥50 nmol/L	23	60.65	889	61.05	34.69 (27.70, 41.68)	1.07	
≥60	<38 μg/day		14	13.94	813	37.90	33.92 (29.18, 38.67)	5.68	*p* = 0.018
	≥38–<75 μg/day		4	50.54	1218	57.54	29.00 (19.21, 38.79)	1.63	
	≥75 μg/day		1	166.67	19	51.75	-	0.89	
		<50 nmol/L	10	17.84	713	29.50	37.66 (33.12, 42.20)	5.77	*p* < 0.001
		≥50 nmol/L	9	41.54	1337	59.27	29.41 (21.88, 36.94)	1.35	

*: Weighted mean (WM) is the product of percentage weighting of studies and corresponding parameter. ƚ Increase in 25(OH)D in nmol/L per 2.5 μg/day (100 IU/day) was calculated as follows: [(Achieved WM 25(OH)D concentration − Baseline WM 25(OH)D concentration)/vitamin D dose μg/day)] × 2.5. *p*-Values were calculated using Mann–Whitney U testing methods.

**Table 4 nutrients-15-03986-t004:** Characteristics among women, dependent on age.

Age in Years	Dose Category	Subgroup	Study Arms, N	Dose, μg/Day *	Vitamin D Group, N	Baseline 25(OH)D, nmol/L *	Weighted Mean Difference in nmol/L (95% CI)	Serum Increment in nmol/L per 2.5 μg/Day Vit.D. ^ƚ^
18–59	All doses	Women	10	38.75	202	33.31	22.36 (13.65, 31.08)	1.41
≥60	All doses	Women	3	10.00	286	35.77	36.46 (26.91, 46.01)	8.42
18–59	<38 μg/day	Women	6	16.50	90	29.84	21.51 (12.21, 30.82)	3.09
≥60	<38 μg/day	Women	3	10	286	35.77	36.46 (26.91, 46.01)	8.42

*: Weighted mean (WM) is the product of percentage weighting of studies and corresponding parameter. ƚ Increase in 25(OH)D in nmol/L per 2.5 μg/day (100 IU/day) was calculated as follows: [(Achieved WM 25(OH)D concentration − Baseline WM 25(OH)D concentration)/vitamin D dose μg/day)] × 2.5.

**Table 5 nutrients-15-03986-t005:** Characteristics of the subgroup ethnic minorities from northern latitudes and northern population.

Age in Years	Dose Category	Subgroup	Study Arms, N	Dose, μg/Day *	Vitamin D Group, N	Baseline 25(OH)D, nmol/L *	Weighted Mean Difference in nmol/L (95% CI)	Serum Increment in nmol/L per 2.5 μg/Day Vit.D. ^ƚ^
18–59	All doses	Ethnic minorities <50 nmol/L baseline	8	17.43	266	20.11	27.62 (22.35, 32.89)	3.65
		Northern population <50 nmol/L baseline	11	78.26	520	37.60	46.94 (28.74, 65.14)	1.69
	<38 μg/day	Ethnic minorities <50 nmol/L baseline	7	16.05	254	19.90	27.11 (21.60, 32.61)	4.01
		Northern population <50 nmol/L baseline	3	11.38	75	32.42	18.36 (4.86, 31.86)	5.73

*: Weighted mean (WM) is the product of percentage weighting of studies and corresponding parameter. ƚ Increase in 25(OH)D in nmol/L per 2.5 μg/day (100 IU/day) was calculated as follows: [(Achieved WM 25(OH)D concentration − Baseline WM 25(OH)D concentration)/vitamin D dose μg/day)] × 2.5.

**Table 6 nutrients-15-03986-t006:** Gender-specific subgroups. No relevant differences in serum 25(OH)D increment per 2.5 μg (100 IU) of vitamin D were observed between genders. No significant gender differences were observed in the weighted mean difference in absolute serum increment. In purely numerical terms, only the absolute but non-significant difference in the weighted mean difference between the >75 μg/day subgroups of men and women is noticeable here. However, this can be explained by the significantly higher dose in the male group.

Age in Years	Dose Category	Subgroup	Study Arms, N	Dose, μg/Day *	Vitamin D Group, N	Baseline 25(OH)D, nmol/L *	Weighted Mean Difference in nmol/L (95% CI)	Serum Increment in nmol/L per 2.5 μg/Day Vit.D. ^ƚ^
18–59	<38 µg/day	Men	6	14.27	167	43.37	23.61 (14.06, 33.15)	3.44
		Women	6	16.37	90	29.42	21.51 (12.21, 30.82)	3.14
	≥38–<75 µg/day	Men	4	63.34	113	50.30	32.78 (23.95, 41.61)	1.53
		Women	2	61.66	94	45.83	29.83 (21.58, 38.08)	1.36
	≥75 µg/day	Men	7	204.10	87	53.55	46.55 (15.54, 77.55)	0.53
		Women	2	89.29	18	32.78	20.72 (−17.11, 58.56)	0.58

*: Weighted mean (WM) is the product of percentage weighting of studies and corresponding parameter. ƚ Increase in 25(OH)D in nmol/L per 2.5 μg/day (100 IU/day) was calculated as follows: [(Achieved WM 25(OH)D concentration − Baseline WM 25(OH)D concentration)/vitamin D dose μg/day)] × 2.5.

**Table 7 nutrients-15-03986-t007:** Vitamin D recommendations.

Subgroup (Study Arms, N)	Vitamin D Recommendation in μg/Day (IU/Day) for 95% of the Population to Reach/Maintain 50 nmol/L ^a^	Vitamin D Recommendation in μg/Day (IU/Day) for 95% of the Population to Reach/Maintain 75 nmol/L ^b^
Overall (n = 73)	24.90 (995.94)	57.26 (2290.48)
18–59 year (n = 54)	35.91 (1436.25)	64.39 (2575.69)
≥60 year (n = 19)	15.49 (619.77)	27.63 (1105.11)

^a^: Formula: [(50 − 5th percentile of Baseline 25(OH)D)/(serum increment in nmol/L/2.5)]; ^b^: formula: [(75 − 5th percentile of Baseline 25(OH)D)/(serum increment in nmol/L/2.5)].

## Data Availability

The datasets used and/or analyzed during the current study are available from the corresponding authors on request.

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
