# Peer review of "Meta-Analysis of European Clinical Trials Characterizing the Healthy-Adult Serum 25-hydroxyvitamin D Response to Vitamin D Supplementation"

_nutrients, 2023, doi:10.3390/nu15183986_

Round 1

Reviewer 1 Report

The authors provide a well structured, elaborate meta-analysis of a very important research question. Especially vitamin D is a widely and sometimes contradictingly discussed supplement, which makes such systematic reviews very valuable. The authors re-evaluate recommendations for oral vitamin D uptake.

The authors fulfill all most important aspects of the PRISMA 2020 Checklist and provide a structured, accurate review.

Minor Issues:

Line 33: please delete one “cardiovascular disease” as it is stated twice.

Line 386: there is a problem with the reference program, given an “Error! Reference source not found”.

Figure S2: The description of the plot is still in German language – please change to English.

Suggestions/Recommendations

The whole work follows the PRISMA guidelines. Since all items of the PRISMA 2020 Checklist for Abstracts are fulfilled except 2, and the information is available in the body text, you might want to add them to the abstract as well:

1.       Item #5 Specify the methods used to assess risk of bias in the included studies.

2.       Item #9 Provide a brief summary of the limitations of the evidence included in the review (e.g. study risk of bias, inconsistency and imprecision).

Reviewer 2 Report

The authors seem to believe that metabolic processes are unaffected by natural selection. If a population lives under conditions that limit vitamin D synthesis—for thousands of years—it will remain as vulnerable to vitamin D deficiency as a population that has not lived with this constraint.

The authors thus argue that “at northern latitudes from about 431 40° N, sunlight is not strong enough to trigger the synthesis of vitamin D in the skin from 432 October to March.” This is true. But it is also true that modern humans have lived at high latitudes for some 30,000 years. During that time, they have adapted to reduced synthesis of vitamin D, just as they have adapted to other aspects of high-latitude environments, such as cold exposure and a high-meat diet (Frost, 2018; Rejnmark et al., 2004; Waiters et al., 1999).

Vitamin D metabolism is subject to natural selection, like any aspect of the human body. If 25(OH)D levels are continually low, over many generations, there will be selection to use vitamin D more efficiently and more sparingly. Such adaptations can take the form of higher uptake of calcium from the intestines, higher rate of conversion of vitamin D to its most active form, stronger binding of vitamin D to carrier proteins in the bloodstream, and greater use of alternative metabolic pathways for calcium uptake (Frost, 2022).

These adaptations to vitamin D scarcity are found not only in populations at high latitudes but also in dark-skinned populations. In the latter case, vitamin D synthesis is reduced by the blocking of UVB by melanin. This has been shown in Americans with varying degrees of African ancestry: both sunlight and diet are 46% less effective in raising the vitamin D level of those individuals who have high African ancestry (Signorello et al., 2010). African Americans thus seem to reach homeostasis for 25(OH)D at a lower level. Indeed, few of them show signs of vitamin D deficiency:

·         African Americans have “a lower prevalence of osteoporosis, a lower incidence of fractures and a higher bone mineral density than white Americans, who generally exhibit a much more favorable vitamin D status” (Robins, 2009). Among women 65 years of age, the risk of a hip fracture by age 80 is only 4% for African Americans versus 11% for European Americans (Harris, 2006).

·         Among teenage girls, calcium retention, bone formation rates, and calcium absorption efficiency are higher in African Americans than in European Americans (Bryant et al., 2003).

·         Among East African immigrant children in Australia, 87% had 25(OH)D levels less than 50 nmol/L and 44% less than 25 nmol/L. None had rickets, the typical result of vitamin D deficiency in children (McGillivray et al., 2007).

·         A review of the literature concludes that “the intake [of calcium] needed to ensure optimal skeletal status is lower in Blacks than in the other racial groups [Europeans and East Asians]” (Hearney, 2002, p. 153).

Homeostasis

Interestingly, the authors found evidence for homeostasis in their study. With increases in vitamin D supplementation, there were correspondingly smaller increases in 25(OH)D levels:

“As shown in Table 3 [as] the vitamin D supplementation dose increased, relative serum 25(OH)D increment per 2.5 µg supplemented vitamin D decreased. In the population ≥60 years of age, this relative serum 25(OH)D increase was more pronounced.” (p. 13)

Supplementation may be unnecessary, and perhaps even harmful, if the body is “pushing back” against it. It should thus be limited to groups, like those over 60 years of age, who show little or no pushback.

This homeostatic “pushback” is perhaps the most interesting finding of this paper. I hope it will inspire further research.

References

Bryant, R.J.; Wastney, M.E.; Martin, B.R.; Wood, O.; McCabe, G.P.; Morshidi, M.; Smith, D.L.; Peacock, M.; Weaver, C.M. Racial differences in bone turnover and calcium metabolism in adolescent females. J. Clin. Endocr. Metab. 2003, 88, 1043-1047. https://doi.org/10.1210/jc.2002-021367  

Frost, P. (2018). To supplement or not to supplement: are Inuit getting enough vitamin D? Études Inuit Studies 40(2): 271-291. https://doi.org/10.7202/1055442ar

Frost P. (2022) The Problem of Vitamin D Scarcity: Cultural and Genetic Solutions by Indigenous Arctic and Tropical Peoples. Nutrients 14(19):4071. https://doi.org/10.3390/nu14194071

Harris, S.S. Vitamin D and African Americans. J. Nutr. 2006, 136, 1126-1129. https://doi.org/10.1093/jn/136.4.1126

Hearney, R.P. Ethnicity, bone status, and the calcium requirement. Nutr. Res. 2002, 22, 153-178. https://doi.org/10.1016/S0271-5317(01)00358-X

McGillivray, G.; Skull, S.A.; Davie, G.; Kofoed, S.E.; Frydenberg, A.; Rice, J.; Cooke, R.; Carapetis, J.R. High prevalence of asymptomatic vitamin-D and iron deficiency in East African immigrant children and adolescents living in a temperate climate. Arch. Dis. Child. 2007, 92, 1088-1093. https://doi.org/10.1136/adc.2006.112813  

Rejnmark, L.; Jørgensen, M.E.; Pedersen, M.B.; Hansen, J.C.; Heickendorff, L.; Lauridsen, A.L.; Mulvad, G.; Siggaard, C.; Skjoldborg, H.; Sørensen, T.B.; et al. Vitamin D insufficiency in Greenlanders on a Westernized fare: ethnic differences in calcitropic hormones between Greenlanders and Danes. Calcified Tissue Int. 2004, 74, 255–263. https://doi.org/10.1007/s00223-003-0110-9  

Robins, A.H. The evolution of light skin color: role of vitamin D disputed. Am. J. Phys. Anthropol. 2009, 139, 447-450. https://doi.org/10.1002/ajpa.21077

Signorello, L.B.; Williams, S.M.; Zheng, W.; Smith, J.R.; Long, J.; Cai, Q.; Hargreaves, M.K.; Hollis, B.W.; Blot, W.J. Blood vitamin D levels in relation to genetic estimation of African ancestry. Cancer Epidem. Biomar. 2010, 19, 2325-31. https://doi.org/10.1158/1055-9965.EPI-10-0482

Waiters, B.; Godel, J.C.; Basu, T.K. Perinatal Vitamin D and Calcium Status of Northern Canadian Mothers and their Newborn Infants. J. Am. Coll. Nutr. 1999, 18, 122-126. https://doi.org/10.1080/07315724.1999.10718839

Minor corrections are needed.
